# Error correction and diversity analysis of population mixtures determined by NGS

Graham R. Wood[1], Nigel J. Burroughs[1], David J. Evans[2] and Eugene V. Ryabov[2]

[1] Warwick Systems Biology Centre, University of Warwick, Coventry, United Kingdom
[2] School of Life Sciences, University of Warwick, Coventry, United Kingdom

## ABSTRACT

The impetus for this work was the need to analyse nucleotide diversity in a viral mix taken from honeybees. The paper has two findings. First, a method for correction of next generation sequencing error in the distribution of nucleotides at a site is developed. Second, a package of methods for assessment of nucleotide diversity is assembled. The error correction method is statistically based and works at the level of the nucleotide distribution rather than the level of individual nucleotides. The method relies on an error model and a sample of known viral genotypes that is used for model calibration. A compendium of existing and new diversity analysis tools is also presented, allowing hypotheses about diversity and mean diversity to be tested and associated confidence intervals to be calculated. The methods are illustrated using honeybee viral samples. Software in both Excel and Matlab and a guide are available at http://www2.warwick.ac.uk/fac/sci/systemsbiology/research/software/, the Warwick University Systems Biology Centre software download site.

Corresponding author
Graham R. Wood,
g.r.wood@warwick.ac.uk

## INTRODUCTION

Next generation sequencing (NGS) is an extremely powerful tool for the analysis of mixed populations in ecology and biology, providing a means to assess community composition (metagenomics) whilst also aiding the discovery of new species (*Radford et al., 2012*). The relatively high error rate of NGS, however, limits the ability to analyze population diversity directly. Population diversity is a crucial ecological measure, relating particularly to selection pressure and (relative) evolutionary fitness. Diversity is critical within biological arenas as well, for instance T cell and viral diversity are key complementary measures of the immune system and infection status respectively. There is thus a pressing need to create tools to appropriately correct NGS error, to estimate diversity introduced by NGS and to estimate inherent virus diversity.

This study was motivated by the prevalence and pathogenicity of honeybee deformed wing virus (DWV) and related viruses. It is known that these may cause both asymptomatic low-level and symptomatic high-level infection in honeybees; the *Varroa* mite is the likely causal factor that produces a shift from the benign to the pathogenic state, correlating with a shift in the level of viral population diversity (*Martin et al., 2012*; *Ryabov*

**Peer**J ______________________________________________

*et al., 2014*). It was the particular need to test for a change in honeybee viral population diversity that stimulated the current work; for this reason, datasets used here relate to honeybee deformed wing virus populations.

The *Varroa* mite is now endemic in honeybee colonies throughout Europe, North America and New Zealand, causing significant colony losses (*Neumann & Carreck, 2010*). *Varroa* acts as a vector for a range of honeybee viral pathogens, most important of which is the picorna-like deformed wing virus, a close relative of *Varroa destructor* virus-1 (VDV-1). At high *Varroa* levels a virus most closely resembling a recombinant between published DWV and VDV-1 sequences is amplified in the colony, leading to overt deformed wing disease and colony loss (*Ryabov et al., 2014*). Thus, to determine the degree of selection for that strain there is a need to compare viral diversity between *Varroa*-free and *Varroa*-infested honeybee viral RNA samples. Viral RNA samples, extracted from individual newly-emerged adult bees from *Varroa*-free or *Varroa*-infested colonies, were subjected to next-generation sequencing (NGS) and the resulting reads used to compare the diversity in the distribution of nucleotides at each nucleotide position, from one sample to the other. Next generation sequencing, however, is subject to error which in turn inflates diversity measures. An accurate measure of nucleotide diversity at each position is needed in this context, and thus the statistical structure of the nucleotide distribution can be used to achieve a means for correction.

An extensive literature on the correction of next generation sequencing error is available, entirely devoted to methods for correction of individual nucleotides rather than the distribution of nucleotides at each position. A useful recent review of the existing literature can be found in *Yang, Chockalingam & Aluru (2013)*, where it is pointed out that all methods depend on "alignment and consensus", with more powerful approaches being developed over time. Methods are classified into three types: *k*-spectrum based (all *k*-mers are identified, then clustered by Hamming distance and clusters assumed to come from the same genomic location), suffix tree/array-based (a generalisation of the *k*-mer approach which accommodates multiple values of *k*) and methods dependent on a multiple sequence alignment. Early methods (*Quince et al., 2009*; *Zagordi et al., 2011*) assumed that errors occur at random while *Skums et al. (2012)* builds in known platform influences. In *Macalalad et al. (2012)* the authors utilised the fact that biological variants and process errors exhibit different covariation. Here we develop a simple but effective method for nucleotide distribution correction. This is a problem at a coarser level than rectification of the assignation of each nucleotide, allowing us to use a different approach. Additionally, the method appears to be unique in that it uses a standard sequence for the purposes of error calibration.

A careful examination of the distribution of Shannon diversity first appeared in the ecology literature (*Hutcheson, 1970*). *Beerenwinkel et al. (2012)* provides an excellent overview of the state-of-the-art in estimating viral diversity from next-generation sequencing data; diversity can be measured at single sites (for single nucleotide variant detection), locally in windows of a multiple sequence alignment and globally over the entire genomic region. In *Gregori et al. (2014)* the authors look at three measures of

diversity and compare their values (the variance of diversity is incorrectly stated in *Gregori et al. (2014)*, but corrected here in the Diversity analysis section). To conclude the review of literature on a broader note, in *Schreiber & Brown (2002)* a method for correcting systematic bias in the nucleotide distribution (the "distortion") in a genome is presented.

The novel stochastic method for NGS error correction of diversity measures introduced here is based on use of an evolutionary model (in the first instance, the Jukes–Cantor model (*Jukes & Cantor, 1969*)) and requires the availability of an accurately known sequenced sample. The corrected nucleotide distribution is used to calculate both the expected diversity and the diversity variance. These in turn can be used to test hypotheses about single diversities, or to test the hypothesis of equality of diversity across two samples. Hypotheses about mean diversities and comparisons of mean diversities of two samples are also discussed, taking into account correlation of diversities across the averaged sites. Finally, the ideas are used to produce a diversity threshold for a sample to be consistent with a clonal (zero nucleotide diversity) population. The methodology is illustrated with NGS datasets drawn from the honeybee study. The methods discussed extend our understanding of NGS error correction and provide a core set of tools useful in the study of diversity.

## MATERIALS AND METHODS

### Error and diversity correction

We first correct the NGS error in the nucleotide distribution and then use this distribution to calculate the (corrected) diversity. Our starting point is the true (to be estimated) nucleotide distribution, $p$. Assuming independence of errors, the error introduced through NGS can be modelled by a $4 \times 4$ matrix $M$ whose entries give the probability of a particular nucleotide change during sequencing, conditional on the initial value of the nucleotide, whence the theoretical distribution of nucleotides following sequencing is $q = Mp$. In practice we sequence finitely many nucleotides $n$ so the empirical distribution is $\hat{q}$ with $E(\hat{q}) = q$. The simplest error model is the Jukes–Cantor model, where a nucleotide mutates to any other with equal (low) probability $\alpha$, summarised as

$$
M = M(\alpha) = \begin{bmatrix} P(A|A) & P(A|C) & P(A|G) & P(A|T) \\ P(C|A) & P(C|C) & P(C|G) & P(C|T) \\ P(G|A) & P(G|C) & P(G|G) & P(G|T) \\ P(T|A) & P(T|C) & P(T|G) & P(T|T) \end{bmatrix}
$$

$$
= \begin{bmatrix} 1-3\alpha & \alpha & \alpha & \alpha \\ \alpha & 1-3\alpha & \alpha & \alpha \\ \alpha & \alpha & 1-3\alpha & \alpha \\ \alpha & \alpha & \alpha & 1-3\alpha \end{bmatrix}.
$$

Since $Mp = p + \alpha(1 - 4p)$, this transformation has the effect of moving all probabilities closer to 0.25, thereby increasing diversity. Richer mutation models can be incorporated, for example allowing different mutation rates between purines and pyrimidines; for this study the Jukes–Cantor model is sufficient for our purposes.

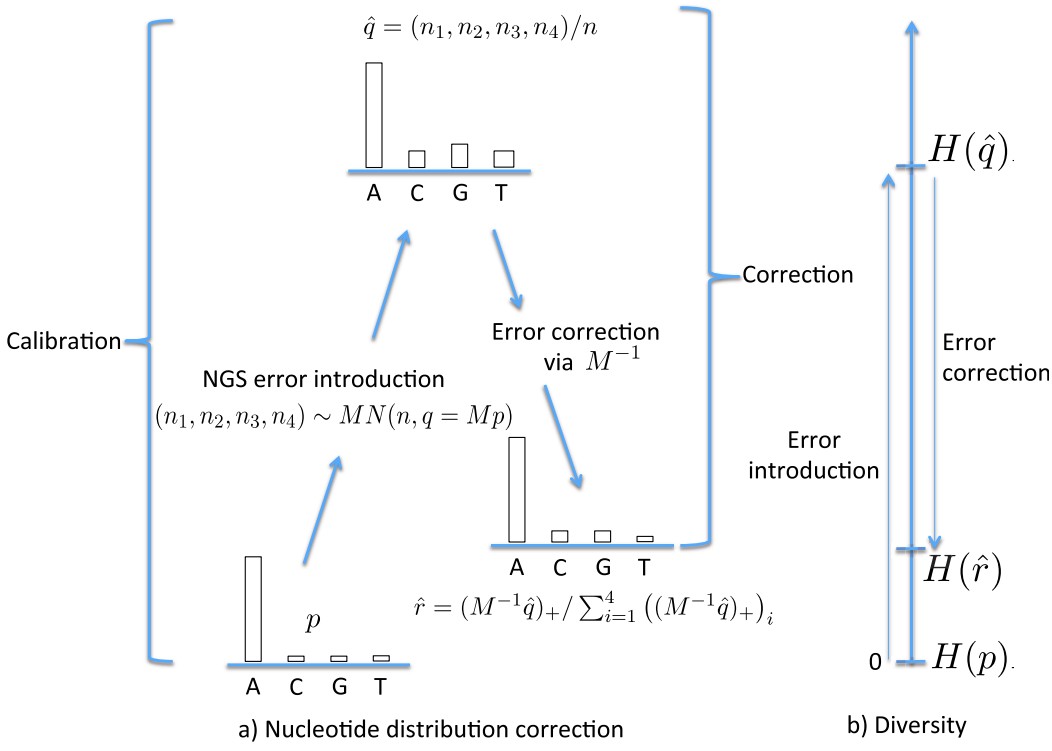

a) Nucleotide distribution correction

b) Diversity

**Figure 1** **The stages of NGS nucleotide distribution error correction.** (A) Error is introduced to the true nucleotide distribution $p$, giving distribution $\hat{q}$, the result of multinomial sampling with $n$ (the coverage, the number of nucleotides sequenced) trials in which the true sequenced nucleotide distribution is given by $q = Mp$. The error is corrected by forming $\hat{r}$, the normalisation of $(M^{-1}\hat{q})_+$. In practice $M$ must be estimated (the calibration step), using a known initial distribution $p$. (B) The initial diversity $H(p)$ increases to $H(\hat{q})$ under NGS, but falls back to $H(\hat{r})$ when corrected.

The NGS error can be partly corrected by reversing the mutation process, first multiplying $\hat{q}$ by $M^{-1}$. In practice, thanks to the sampling variation introduced during NGS, the reversal can yield negative components. For this reason we must bring these back to zero and then normalize so that the nucleotide proportions sum to one. Thus, the corrected nucleotide distribution $\hat{r}$ is the component-wise maximum of $M^{-1}(\hat{q})$ and $(0, 0, 0, 0)$, denoted $(M^{-1}\hat{q})_+$, normalised, so $\hat{r} = (M^{-1}\hat{q})_+ / \sum_{i=1}^{4} ((M^{-1}\hat{q})_+)_i$. Since $M^{-1}(\hat{q}) = (q - \alpha)/(1 - 4\alpha)$, this transformation has the effect of moving all probabilities away from 0.25, so decreasing diversity. Only when $\hat{q} = Mp$ does this reversal process work exactly, returning $p$. Figure 1A lays out these stages, from $p$, the true nucleotide distribution in the organism, to $\hat{q}$ the distribution following NGS, to $\hat{r}$, the corrected distribution. To illustrate this numerically, with $p = (1, 0, 0, 0)$, $\alpha = 0.001$ and $n = 10,000$ then $(n_1, n_2, n_3, n_4)$ could be $(9989, 4, 3, 4)$ (using a multinomial distribution) whence $\hat{q} = (0.9989, 0.0004, 0.0003, 0.0004)$, $M^{-1}\hat{q} = (1.0019, -0.0006, -0.0007, -0.0006)$, $(M^{-1}\hat{q})_+ = (1.0019, 0, 0, 0)$, and $\hat{r} = (1, 0, 0, 0) = p$. In this example the correction is exact, but this is not always the case, whence the estimate $\hat{r}$ will still be subject to error.

To utilize this correction method, we need the matrix $M$; in the case of the Jukes–Cantor model this requires an estimate of the mutation rate $\alpha$. This can be achieved by estimating
it on a sample of known composition and genotype, denoted here the "standard" sample. This provides a two-step method to correct an empirical nucleotide distribution $\hat{q}$ for NGS error. The first (calibration) step uses $p_{standard}$ and $\hat{q}_{standard}$ to estimate $\alpha$ (as $\hat{\alpha}$, so giving an estimate $\hat{M}$ of $M$) while the second (correction) step uses $\hat{M}$ and $\hat{q}$ to estimate $p$ with $\hat{r}$, as follows:

1. Calibration Step. This is the estimation of $\alpha$, the error rate in NGS sequencing. We begin with a sample in which the nucleotide mix $p_{standard} = (p_1, p_2, p_3, p_4)^T$ is known at each nucleotide position along a genome. We then sequence the mix using NGS, giving nucleotide mix $\hat{q}_{standard} = (\hat{q}_1, \hat{q}_2, \hat{q}_3, \hat{q}_4)^T$. For each site we can solve for $\hat{\alpha}$ using $\hat{q} = \hat{M}p$. Equating $i$th components provides estimates $\hat{\alpha}_i = (\hat{q}_i - p_i)/(1 - 4p_i)$ for $i = 1, \ldots, 4$; these can be averaged across nucleotides to give a lower variance estimate of $\alpha$,

$$\hat{\alpha} = \frac{\hat{\alpha}_1 + \hat{\alpha}_2 + \hat{\alpha}_3 + \hat{\alpha}_4}{4} = \frac{1}{4}\sum_{i=1}^{4}\frac{\hat{q}_i - p_i}{1 - 4p_i}. \tag{1}$$

These estimates can be averaged across nucleotide positions to improve the estimate further.

2. Correction Step. Given nucleotide distributions $\hat{q}$ produced from NGS of a sample, form the corrected counts $\hat{r} = (\hat{M}^{-1}\hat{q})_+ / \sum_{i=1}^{4}\left((\hat{M}^{-1}\hat{q})_+\right)_i$, where $\hat{M} = M(\hat{\alpha})$.

Example 1 in the Results section illustrates these calibration and correction steps with a numerical example.

It is possible to estimate an upper bound for the residual error rate remaining after correction, given an NGS error rate $\alpha$ and coverage $n$. This is done by using the calibration step with $p_{standard} = (1, 0, 0, 0)$ and $\hat{q}_{standard} = \hat{r}$, denoting the estimated residual error rate by $\hat{\beta}$. Here $\hat{r}$ is found by first generating $n$ values using $q = M(\alpha)p$, then correcting using rate $\alpha$. A graph of $\hat{\beta}$ against $\alpha$ is given in Fig. 2, for the case $p_{standard} = (1, 0, 0, 0)$ and $n = 1,000, 2,000, 5,000, 10,000$ and $20,000$. This demonstrates, for these parameters, that correction reduces the error by a factor of over 10 for a coverage of $n = 1,000$. That this is an upper bound is made clear in the later Discussion section.

Given a nucleotide probability mass distribution $p = (p_1, p_2, p_3, p_4)$ the nucleotide diversity is $H = -\sum_{i=1}^{4}p_i\log_4 p_i$. This can be thought of as the geometric mean $\prod_i p_i^{p_i}$ of the distribution probability masses, transformed by taking the natural logarithm of the reciprocal; then no diversity, for example (1,0,0,0), has $H = 0$ while complete diversity, a uniform distribution of (0.25, 0.25, 0.25, 0.25), has $H = 1$. Figure 3 shows the graph of a component $-x\log_4 x$ of diversity. The steep slope at the extremes, particularly at zero, shows that correction to a component at these extremes has most effect on diversity measurement. Of interest here is detection of clonality, where $H = 0$, so correction of very small probabilities is critical.

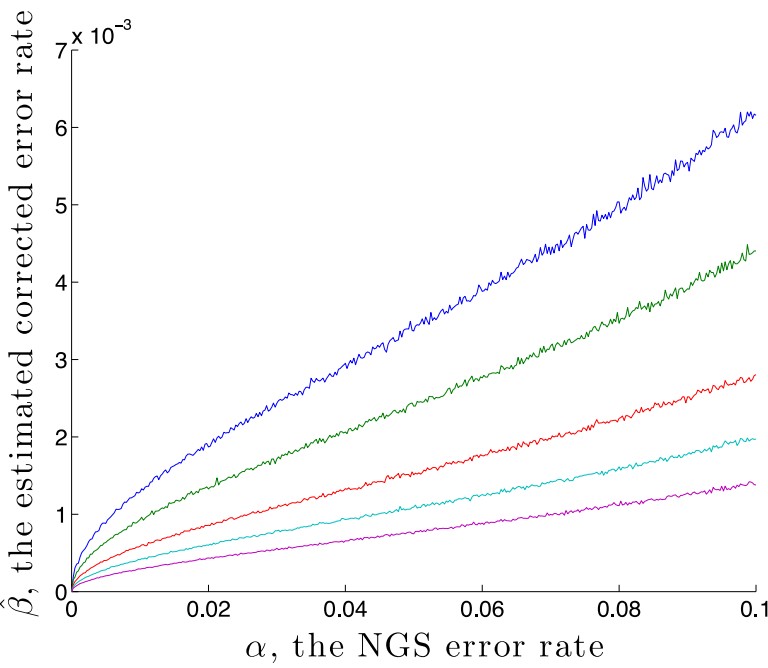

**Figure 2** **The estimated residual error rate $\hat{\beta}$ after correction plotted against the initial error rate $\alpha$, when the true initial distribution is the degenerate $p = (1, 0, 0, 0)$.** The larger the coverage $n$, the more accurately $\hat{q}$ estimates $q$ and so the more dependable the correction, hence the smaller is $\beta$. Curves are shown for coverage $n$ of 1,000 (blue), 2,000 (green), 5,000 (red), 10,000 (cyan) and 20,000 (magenta). Each point on the graph is the mean of $N = 5,000$ replicate trials.

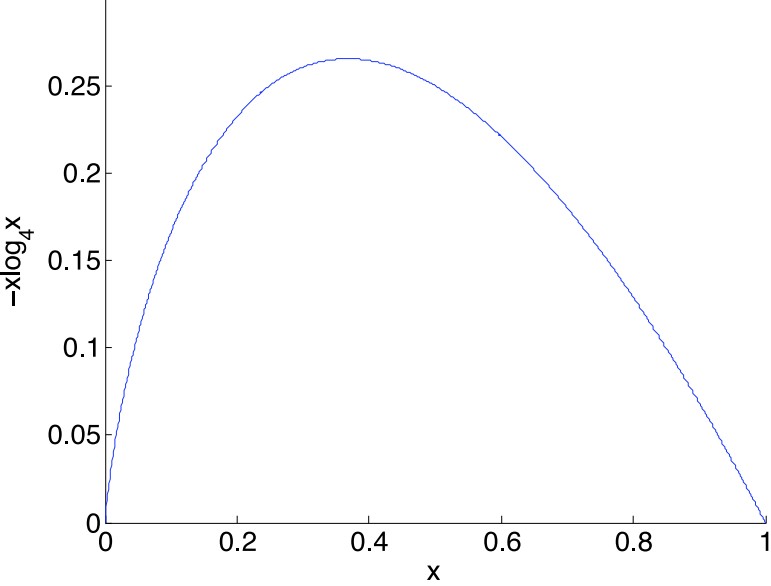

**Figure 3** **The graph of a component of diversity, $-x\log_4 x$.** The gradient converges to infinity as $x$ goes to zero. A consequence is that corrections in this region have a large influence on diversity measurement.

With the corrected nucleotide distribution available, the Shannon diversity can be estimated as

$$\hat{H} = H(\hat{r}) = -\sum_{i=1}^{4} \hat{r}_i \log_4 \hat{r}_i. \tag{2}$$

Figure 1B schematically shows the relative levels of $H(p)$ (true diversity), $H(\hat{q})$ (uncorrected diversity) and $H(\hat{r})$ (corrected diversity). We conclude this section by remarking that the data used for calibration should ideally be sequenced under similar conditions to the data which is to be corrected.

## Diversity analysis

Two questions are key in an analysis of population diversity. Firstly, whether the diversity $H$ is equal to a given value $\mathscr{H}$ and secondly, whether two populations have the same diversity, or alternatively whether one is more diverse than the other. Specifically, we want to test null hypotheses, firstly of the form $H_0 \colon H = \mathscr{H}$ and secondly of the form $H_0 \colon H_1 = H_2$, for two samples labeled '1' and '2'. We will examine both of these tests for diversity measured at a single nucleotide position and for the mean diversity across segments comprising multiple $(N)$ possibly correlated sites.

In all cases we will have one or more true nucleotide distributions $p$, true post NGS nucleotide distributions $q = Mp$, estimates $\hat{q}$ determined using a sample of $n$ nucleotides drawn from $q$, so $\hat{q}_i = n_i/n$ for each $i = 1, \ldots, 4$, where $n_i$ is the number of draws resulting in the $i$th nucleotide and finally, corrected distributions $\hat{r}$. Given $\hat{r}$, $\hat{H} = H(\hat{r})$, as in (2), estimates the true diversity $H = -\sum_i p_i \log_4 p_i$ with $H$ evaluated at corrected values $\hat{r}$ of estimates $\hat{q}$ of true values after error introduction, $q$. When $n$ is large, this approximation will be close to the true value. In general, since the diversity function is concave, diversity is increased when we replace $p$ with $q$ and decreased when we replace $q$ with $\hat{q}$ (since $\hat{H}$ is known to underestimate $H$ (Hutcheson, 1970)) and then $\hat{r}$ (since $\hat{r}$ corrects $\hat{q}$). This compensation is fortuitous.

The variance of Shannon diversity $\mathrm{Var}(\hat{H})$ at a nucleotide position with coverage $n$ and empirical distribution $\hat{r} = (\hat{r}_1, \hat{r}_2, \hat{r}_3, \hat{r}_4)$ is approximated to first order (Hutcheson, 1970) with the following expression,

$$\mathrm{Var}(\hat{H}) \approx \frac{1}{n}\left(\sum_{i=1}^{4} \hat{r}_i(\log_4 \hat{r}_i)^2 - \left(\sum_{i=1}^{4} \hat{r}_i \log_4 \hat{r}_i\right)^2\right). \tag{3}$$

This can be shown using the $\Delta$-method and that $\mathrm{Var}(\hat{r}_i) = r_i(1 - r_i)/n$ and $\mathrm{Cov}(\hat{r}_i \hat{r}_j) = -r_i r_j/n$, where $E(\hat{r}_i) = r_i$ for $i = 1, \ldots, 4$.

For testing and estimation we also need the variance of the mean of diversities averaged across a segment of sites. For ease of presentation (though it is readily generalised), we now assume that (3) provides the variance of estimated diversity at each nucleotide position, denoted $s^2$. Empirical evidence shows that diversity is correlated across positions. We assume an exponential decay of the diversity correlation, with correlation between

**Table 1 Test statistics and two-sided 95% confidence intervals, in brackets, for diversity testing and estimation with a single position and one or two samples (columns).** A population mean is denoted by $\mu$. For the corresponding formulae involving means across multiple positions, $\bar{H}$ replaces $\hat{H}$.

| | Number of samples | | |
|---|---|---|---|
| | **1** | | **2** |
| **Hypothesis** | **Test statistic $z$** | **Hypothesis** | **Test statistic $z$** |
| $\mu_H = \mathscr{H}$ | $\dfrac{\hat{H} - \mathscr{H}}{\sqrt{\mathrm{Var}(\hat{H})}}$ | $\mu_{H_1} - \mu_{H_2} = 0$ | $\dfrac{\hat{H}_1 - \hat{H}_2}{\sqrt{\mathrm{Var}(\hat{H}_1) + \mathrm{Var}(\hat{H}_2)}}$ |
| | $\left[\hat{H} \pm z_{0.975}\sqrt{\mathrm{Var}(\hat{H})}\right]$ | | $\left[\hat{H}_1 - \hat{H}_2 \pm z_{0.975}\sqrt{\mathrm{Var}(\hat{H}_1) + \mathrm{Var}(\hat{H}_2)}\right]$ |

nucleotide positions $i$ and $j$ given by $\rho^{j-i+1}$. Then the average diversity across $N$ consecutive nucleotide positions, denoted $\bar{H}$, has variance $\mathrm{Var}(\bar{H})$ given by

$$\mathrm{Var}(\bar{H}) \approx \frac{s^2}{N}\left(1 + 2\frac{N-1}{N}\frac{\rho}{1-\rho} - \frac{2}{N^2}\left(\frac{\rho}{1-\rho}\right)^2(1-\rho^{N-1})\right) \tag{4}$$

$$= \frac{s^2}{N}\left(1 + \frac{2\rho}{1-\rho}\right) \quad \text{for } N \text{ large.} \tag{5}$$

We estimate $\rho$ using the correlation between adjacent diversities (the one-step lagged correlation) along the segment of interest. With these variance estimates we can conduct hypothesis tests and construct confidence intervals in all cases considered. These are summarised in Table 1 and illustrated numerically in Example 2 in "Results".

## Clonal threshold estimation

Suppose that a clonal sample (one with zero diversity) is sequenced by next generation methods with error rate $\alpha$. This is corrected to level $\hat{\beta}$, as shown in Fig. 2. This level of error remains in the sample, whence an upper 95% threshold for $\bar{H}$, the average diversity across $N$ sequential positions, with lagged correlation $\rho$, is

$$\bar{H} + z_{0.95}\sqrt{\mathrm{Var}(\bar{H})}$$

where $z_{0.95} = 1.64$ is the 95th percentile of the standard normal distribution. Note that this threshold depends on both the coverage $n$ and the segment length $N$. A larger $n$ typically decreases $\bar{H}$ and so the threshold level, while a larger $N$ decreases the width of the confidence interval around $\bar{H}$, so also decreases the threshold level.

## Sequence data

Three viral samples were sequenced by next generation sequencing (Illumina HiSeq1000). These comprised a sample that could be used for estimation of the error (standard sample), and two experimental samples, from low viral load and high viral load bees. After trimming adapters and barcodes, the first reads (the reads are paired), 101nts long, were used to calculate the NGS error rate and illustrate the methods.

### Standard sample: an accurately known mix of two viral recombinants, X59

This was composed of two of the DWV-like viral recombinant RNA genomes described in *Moore et al. (2011)*, VDV-1$_{DVD}$ and VDV-1$_{VVD}$ (NCBI Accession Nos. HM067437 and HM067438 respectively). The mixture was produced by *in vitro* RNA transcription using linearized plasmid clones with full-length cDNA inserts of VDV-1$_{DVD}$ and VDV-1$_{VVD}$ (*Moore et al., 2011*) with the T7 mMESSAGEmMACHINE kit (Ambion). The RNA transcripts were purified using RNAeasy columns (Quiagen), quantified, mixed 25% VDV-1$_{DVD}$, 75% VDV-1$_{VVD}$ and sequenced using the Illumina platform protocol. The NGS reads are available in the EBI Sequence Read Archive, study accession PRJEB5249, ERS395188. The reads were aligned using Bowtie 2 (*Langmead & Salzberg, 2012*) to a single reference sequence (VDV-1$_{DVD}$). SAMtools mpileup (*Li et al., 2009*) was used to produce the number of nucleotides covering each position in the reference. We excluded positions of mismatches between the VDV-1$_{DVD}$ and VDV-1$_{VVD}$ sequences, about 4% of the genome length. This produced nucleotide pileups where all diversity was a result of the methods used during NGS sequence acquisition. This dataset is used for calibration of NGS error.

### A viral population mix from low DWV level (F3) and from high DWV level (E7) honeybees

Samples from honeybee nurse bees for which qRT-PCR showed either low DWV levels (EBI SRA study accession PRJEB5249, ERS395182) or high DWV levels (EBI SRA study accession PRJEB5249, ERS395180) were used. In each case, in order to assemble the required nucleotide distribution data, the following computational steps were carried out. "First reads" of length 101nt were aligned using Bowtie 2 to DWV (NC_004830) and VDV-1 (NC_006494) reference sequences. SAMtools mpileup was then used to produce the number of nucleotides covering each position in the references.

## Deformed wing virus

A map of the deformed wing virus adapted from *Lanzi et al. (2006)* is shown in Fig. 4. In the Results section we select a single site in the helicase region of the genome (Examples 2.1 and 2.2), and average across the capsid region (Examples 2.3 and 2.4).

## RESULTS

**Example 1. Calibration and error correction**

**(i) Calibration—estimation of the NGS error rate α.** Nucleotide pileups resulting from the X59 mix of two viral recombinants were used for calibration of the NGS error rate. Nucleotide positions where DWV and VDV-1 are identical were selected; for these the true distribution is one of $p = (1, 0, 0, 0)$, $(0, 1, 0, 0)$, $(0, 0, 1, 0)$ or $(0, 0, 0, 1)$. For example, the upper half of Table 2 shows the pileup counts for the first three capsid (see Fig. 4) nucleotide sites. Since the $p$ here place all weight on one nucleotide, Eq. (1) reduces, when $p = (1, 0, 0, 0)$, for example, to $(\hat{q}_2 + \hat{q}_3 + \hat{q}_4)/3$ or the average error proportion. Thus, for the first position in Table 2 we would estimate $\alpha$ as $(0.0010 + 0.0016 + 0.0008)/3$ or 0.0011; for the second and third positions the corresponding averages are 0.0014 and

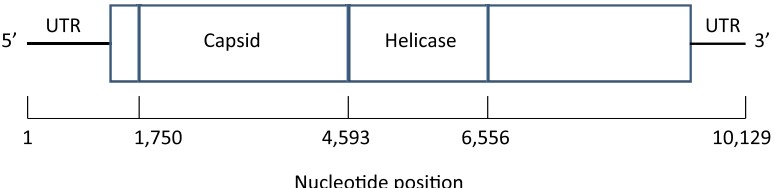

**Figure 4 Map of the genome of the deformed wing virus, a positive-stranded RNA virus.** The capsid region codes for the four structural proteins, multiple copies of which are used to build the icosahedral virus shell, while the helicase region is involved in the unfolding of the RNA strand. Untranslated regions (UTRs) flank the coding sequence.

**Table 2 Correction of NGS errors.** The raw nucleotide counts, the relative frequency distribution $\hat{q}$ of nucleotides after next generation sequencing and the relative frequency distribution after correction $\hat{r}$ are shown for the first three X59 dataset capsid positions (upper half of table); here calibration and correction are both based on X59. For each position the correct distribution of bases was recovered. The corresponding data and corrections for dataset E7 (a dataset independent of the calibration dataset X59) are shown in the lower half of the table, the method again yielding complete correction.

| Position no. | | A | C | G | T | Diversity |
|---|---|---|---|---|---|---|
| 1 | Counts $\{n_i\}$ | 87,573 | 86 | 142 | 73 | |
| | Proportion $\hat{q}$ | 0.9966 | 0.0010 | 0.0016 | 0.0008 | 0.0222 |
| | Proportion $\hat{r}$ | 1 | 0 | 0 | 0 | 0 |
| 2 | Counts $\{n_i\}$ | 90 | 130 | 156 | 86,961 | |
| | Proportion $\hat{q}$ | 0.0010 | 0.0015 | 0.0018 | 0.9957 | 0.0270 |
| | Proportion $\hat{r}$ | 0 | 0 | 0 | 1 | 0 |
| 3 | Counts $\{n_i\}$ | 68 | 29 | 87,998 | 93 | |
| | Proportion $\hat{q}$ | 0.0008 | 0.0003 | 0.9978 | 0.0010 | 0.0149 |
| | Proportion $\hat{r}$ | 0 | 0 | 1 | 0 | 0 |
| 1 | Counts $\{n_i\}$ | 13,267 | 11 | 16 | 11 | |
| | Proportion $\hat{q}$ | 0.9971 | 0.0008 | 0.0012 | 0.0008 | 0.0161 |
| | Proportion $\hat{r}$ | 1 | 0 | 0 | 0 | 0 |
| 2 | Counts $\{n_i\}$ | 11 | 25 | 15 | 13,215 | |
| | Proportion $\hat{q}$ | 0.0008 | 0.0019 | 0.0011 | 0.9962 | 0.0208 |
| | Proportion $\hat{r}$ | 0 | 0 | 0 | 1 | 0 |
| 3 | Counts $\{n_i\}$ | 8 | 9 | 13,440 | 10 | |
| | Proportion $\hat{q}$ | 0.0006 | 0.0007 | 0.9980 | 0.0007 | 0.0120 |
| | Proportion $\hat{r}$ | 0 | 0 | 1 | 0 | 0 |

0.0007, giving a local average of 0.0011. This was done for the entire capsid region of the genome (2,843nts in length), yielding $\hat{\alpha} = 0.001949$, so the overall NGS error rate is estimated at a little under 2 in 1,000.

**(ii) Correction of the distributions $\hat{q}$.** This was carried out in three ways.

(1) Within the capsid region of X59, using $\hat{\alpha} = 0.001949$ calculated using the entire capsid region. Correction of the distribution $\hat{q}$ in the first three capsid positions, shown in the upper half of Table 2, yielded the $\hat{r}$ distributions shown; each is corrected to the exact distribution. (The Excel software available was used to also

test the calibration-correction routine across a central region in the helicase and a non-structural region downstream of the helicase; in both cases the degenerate distributions were found with excellent accuracy.)

(2) Within the capsid region of E7, again using $\hat{\alpha} = 0.001949$ calculated using the entire capsid region of the independent standard sample X59. The accurate results are shown in the lower half of Table 2.

(3) Within the capsid region of X59, using cross-validation. The capsid region was divided into sites with odd index (for calibration) and sites with even index (for correction). The calibration calculation here gives $\hat{\alpha} = 0.001582$. Two arbitrarily chosen even indices in the correction region (where the true distribution is known to be degenerate) had associated raw nucleotide counts $\{n_i\}$ of $(71, 91461, 39, 36)$ (so $\hat{q} = (0.0008, 0.9984, 0.0004, 0.0004)$) and $(134, 60, 90, 65615)$ (so $\hat{q} = (0.0020, 0.0009, 0.0014, 0.9957)$). The corrected distributions $\hat{r}$ are $(0, 1, 0, 0)$ and $(0.0005, 0, 0, 0.9995)$ respectively, showing considerable improvement.

### Example 2: Diversity testing and confidence intervals

#### 2.1 Single site, single population

The nucleotide distribution, in A, C, G, T order, at arbitrarily chosen position 5201 (within the helicase region of the viral genome, Fig. 4) in F3 is $(1, 0, 58, 1)$ giving $\hat{q} = (0.0167, 0, 0.9667, 0.0167)$. Correction using estimate $\hat{\alpha} = 0.001949$ yields corrected proportions $\hat{r} = (0.0148, 0, 0.9704, 0.0148)$. True diversity is then estimated as $H(\hat{r}) = 0.1110$ using (2) and $\text{Var}(\hat{H}) = 0.0044$ using (3), whence the test statistic $z$ is 1.6814 with $p$-value of 0.0463, leading to rejection of the null hypothesis that the true diversity $\mathscr{H} = 0$ at the 5% level. A 95% confidence interval for true diversity is $[0, 0.2404]$.

Sample E7 at nucleotide position 5201 has counts of $(38, 35, 26497, 48)$ giving $\hat{q} = (0.0014, 0.0013, 0.9954, 0.0018)$ and $\hat{r} = (0, 0, 1, 0)$. True diversity is estimated as $\hat{H} = 0$ and $\text{Var}(\hat{H}) = 0$ (since (3) converges to zero in this case), whence the hypothesis of zero diversity is not rejected. A 95% confidence interval for true diversity is $\{0\}$.

#### 2.2 Single site, two populations

We test whether the diversities are equal at nucleotide position 5201 for F3 (first sample) and E7 (second sample), using the data in Example 2.1. The test statistic $z$ described in Table 1 is 1.6814 with $p$-value $= 0.0463$. Hence we reject the null hypothesis of equal true diversities for the two populations at this site, concluding that F3 has higher diversity. A 95% confidence interval for the true diversity difference is $[-0.0184, 0.2404]$.

#### 2.3 Multiple sites, single population

We test whether the mean diversity across the capsid region (see Fig. 4) of the viral genome is zero, for both E7 and F3 ($N = 2,843$). For E7, $\bar{H} = 0.0037$, the serial correlation of diversities is estimated as $\hat{\rho} = 0.1728$ and mean site diversity variance is $s^2 = 7.714 \times 10^{-7}$ whence $\text{Var}(\bar{H}) = 3.847 \times 10^{-10}$ using (5), giving $z = 187.5$ so that the $p$-value is negligible. Here the sample size is so large and hence the power so great that we conclude that the true diversity is non-zero. Of greater interest is a 95% confidence

interval for the true diversity, namely $[0.0036, 0.0037]$. Corresponding calculations for the more varied F3 sample give values of $\bar{H} = 0.0410$, $\hat{\rho} = 0.1195$, $s^2 = 0.0019$, $\text{Var}(\bar{H}) = 8.702 \times 10^{-7}$, $z = 43.98$ and again a negligible $p$-value. A 95% confidence interval for the true mean diversity value is $[0.0392, 0.0429]$.

**2.4 Multiple sites, two populations**

We test the null hypothesis that the true mean diversities, over the capsid region, of F3 (first sample) and E7 (second sample) are equal. Using values already given in Example 2.3, we find that $z = 40.03$, again giving a negligible $p$-value. Hence we conclude that the mean diversity of F3 is greater than that of E7. A 95% confidence interval for the true mean diversity difference is $[0.0355, 0.0392]$.

**Example 3: Clonal threshold**

Using $\hat{\alpha} = 0.001949$ and with coverage of $n = 20,000$ we find that $\hat{\beta} = 0.0001233$. This gives $H(\hat{\beta}) = 0.002748$, by direct calculation using $\hat{r} = -\log_4(1 - 3\hat{\beta})^{1-3\hat{\beta}} - 3\log_4\hat{\beta}^{\hat{\beta}}$. Then $s = \sqrt{\text{Var}(H(\hat{\beta}))} = 0.0008940$ using (3). The 95% clonal diversity threshold for a single site is therefore $0.002748 + 1.64 \times 0.0008940$ or approximately 0.0042. For mean diversity, with $N = 2,843$ and $\hat{\rho} = 0.1728$ (appropriate for E7 on the capsid) the 95% clonal mean diversity threshold is $H(\hat{\beta}) + 1.64(s/\sqrt{N})\sqrt{1 - 2\hat{\rho}/(1 - \hat{\rho})} = 0.0028$.

Figure 5 illustrates the confidence intervals found in Example 2.3, labelled "After correction", together with corresponding confidence intervals for the uncorrected data, labelled "Before correction". The corrected clonal mean diversity threshold of Example 3 is also shown. The highly significant difference between the *Varroa*-free and *Varroa*-infested nucleotide diversities are evident. Correction of the low diversity sample has a far greater effect on diversity than correction of the high diversity sample, due to the steep slope of the diversity component near zero (Fig. 3). The corrected low diversity sample lies just above the clonal threshold.

## DISCUSSION

Here we have proposed a simple method for correction of nucleotide distributions containing errors arising from NGS. The correction method assumes an independent site error model. Throughout we have used the simplest error matrix $M$, the Jukes–Cantor matrix, with the error rate assumed to be constant across the genome and mutation parameter fitted from a standard (known genotype) sample. The method could be extended to use a richer evolutionary model, with parameters tailored to regions of the genome. For instance, the two-parameter Kimura model assumes transversion and transition errors differ, while the Generalized Kimura model adds a further parameter, allowing transversion error probabilities to differ according to transversion direction (*Ewens & Grant, 2005*). At an extreme, a full 12-parameter model could be used, fitted in sets of three consecutive nucleotides, since four linear equations are available per position. In principle, both the statistical nature of the error mechanism of the NGS platform and errors introduced in sample generation (e.g., PCR) should be captured in the model used.

A geometric view of the correction process is shown in Fig. 6. The tetrahedron is the locus of all probability mass functions on the four points $\{A, C, G, T\}$, with the vertices

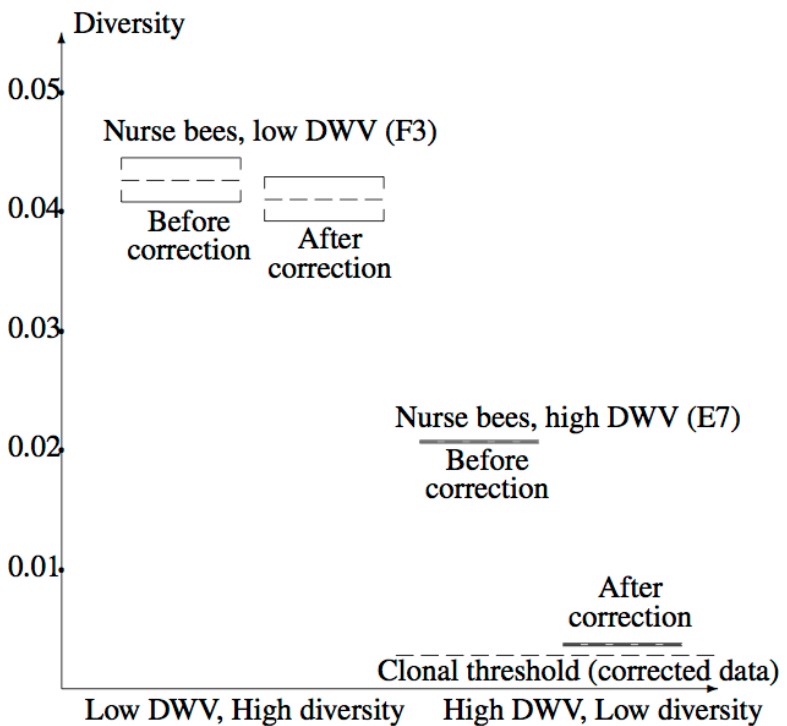

**Figure 5 Confidence intervals for true mean diversity across the capsid region, for the high diversity (F3) and low diversity (E7) samples, both before and after correction for the NGS error.** Correction has a larger effect when the (uncorrected) diversity is low, clearly revealing the reduction in diversity from F3 (low DWV level) to E7 (high DWV level). The clonal threshold for the mean shows that the corrected E7 data plausibly has non-zero diversity.

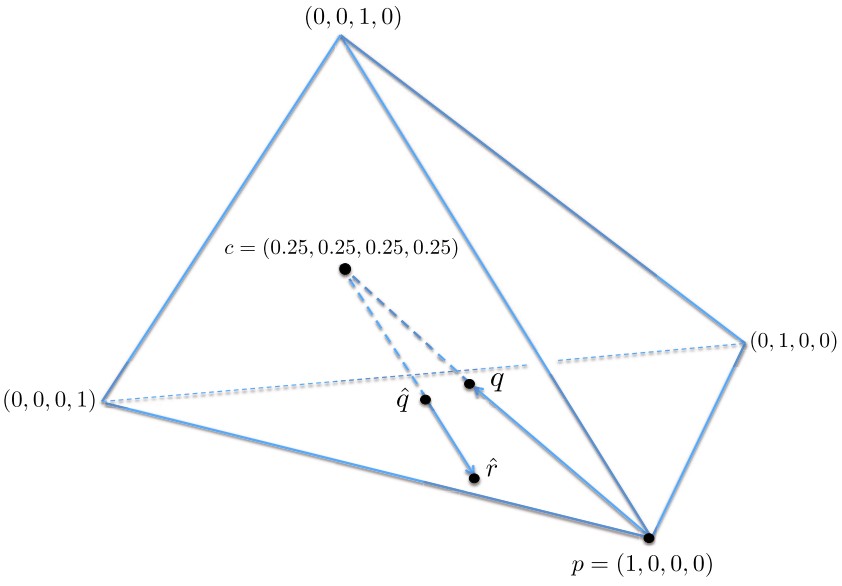

**Figure 6 A geometric view of the correction process, showing the tetrahedron of all nucleotide distributions.** When the true nucleotide distribution $p$ is $(1, 0, 0, 0)$, the limiting distribution after next generation sequencing $q$ is along the line from $p$ to the centroid $c$. Under sampling, we see nearby $\hat{q}$ which is corrected to $\hat{r}$, lying on the extension of the line from $c$ to $\hat{q}$.

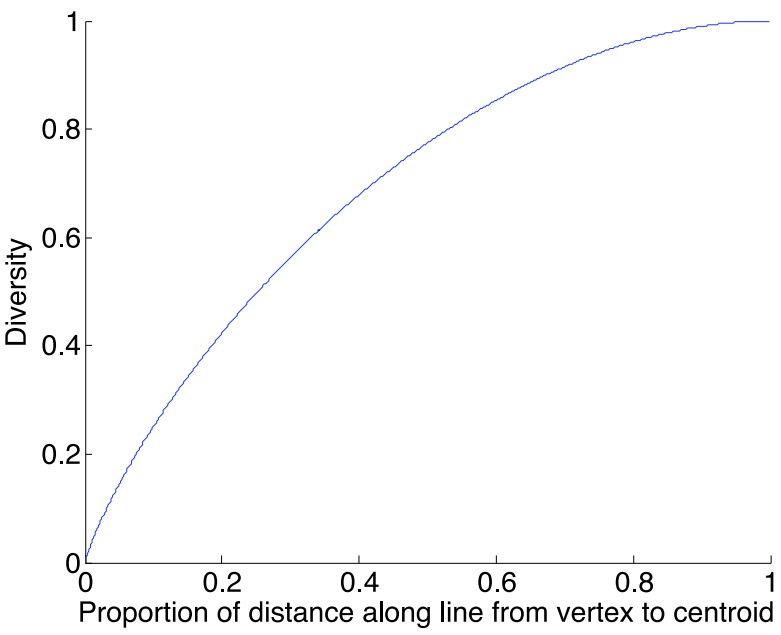

**Figure 7 The diversity profile, from vertex to centroid.** The increase is greatest at the vertex and least at the centroid.

corresponding to situations where the probability mass is on a single nucleotide and the diversity is zero. The centroid $c = (0.25, 0.25, 0.25, 0.25)$ corresponds to the situation where mass is spread equally across nucleotides, and the diversity takes its maximum value of one. When the true nucleotide distribution is $p$ (e.g., $p = (1, 0, 0, 0)$ in Fig. 6) and the NGS error rate is $\alpha$, the limiting distribution following NGS (based on sequencing a very large number of nucleotides) is $q = Mp = (1 - 4\alpha)p + 4\alpha c$, along the line between $p$ and $c$. In practice we sequence finitely many nucleotides $n$, giving an empirical distribution $\hat{q}$ close to $q$. The correction of $\hat{q}$, $\hat{r} = (\hat{q} - 4\alpha c)/(1 - 4\alpha)$, reverses the $p$ to $q$ contraction with $\hat{r}$ lying along the extension of the line from $c$ to $\hat{q}$. (If this $\hat{r}$ lies outside the simplex, $\hat{r}$ is taken to be the nearest point in the simplex to its projection onto the plane determined by its non-negative components.) In general, the further $p$ is from $c$, the greater the distance from $p$ to $q$, so the greater the distance from $\hat{q}$ to $\hat{r}$; the greater the error, the greater the correction. In turn, diversity increases from zero to one along the line from a tetrahedron vertex to the centroid, and is convex down, as shown in Fig. 7, so changes more quickly when the diversity is low and is relatively stable when diversity is high. In summary,

 (i)  The method corrects more when more correction is called for, and

(ii)  The method corrects more where the benefit of correction is greater.

The upshot is that bias inherent in the uncorrected diversity values is rectified by the correction process.

   We caution that error remains, even after correction. In the light of the geometry, worst-case remaining error occurs for $p$ a vertex. In this case Fig. 2 provides information about the worst-case error rate $\hat{\beta}$ remaining after correction. This decreases as $\alpha$ decreases

(since lower $\alpha$ introduces lower error) and decreases as the coverage $n$ increases (since as $n$ increases, $\hat{q}$ becomes closer to $q$).

The diversity testing and estimation stage uses a diversity correlation structure. We have assumed serial correlation of diversities, since this was seen in the samples considered. Other correlation structures could be employed, such as a structure that decays more rapidly with separation.

The method is able to correct in multi-genome situations which appear to lie beyond the reach of other methods. All correction methods in the literature known to the authors are variants of alignment of reads to a reference genome and the taking of a consensus. If the original sample is a mix of two genomes then such methods are unable to correct at sites of variation (since the minor variant will generally be corrected to the major variant), whereas the proposed method estimates the error rate (using a calibration sample) and so is able to correct, even at sites of variation.

Finally, it may be possible to use the "clonal threshold" to detect SNPs or the more general SNVs. This is a topic for future research.

## CONCLUSION

Movement of a virus to a new host can trigger changes in the level of diversity of variants of the virus. Such diversity changes can be detected using next generation sequencing of the viral mix, but error introduced in the process may mask a diversity change. We have presented a method for estimating measurement-error-corrected diversity from NGS data, both at single nucleotide sites and along sections of a genome. A simple but effective method is used to correct for NGS errors in the nucleotide distribution. With these corrected estimates we are able to compare diversity between samples, assessing whether they are consistent with equal diversity, and also assess whether a diversity estimate is consistent with a clonal population. We demonstrated our method within the context of viral diversity in bees that either have low viral load or high viral load, showing that their viral diversity is significantly different and that under high viral load the viral population is near-clonal. Correction for NGS error was particularly important when applied to the sample with low diversity, since the diversity estimate decreased by a factor of more than five. This analysis has demonstrated that the *Varroa* mite has a significant effect on the DWV population in honeybees, due to the emergence of a virulent strain that is preferentially replicated after transmission by *Varroa*. This results in the striking reduction in viral diversity observed in *Varroa*-exposed pupae and symptomatic worker bees. Our method is applicable to any population structure analysis using NGS data and thus adds a valuable tool to the study of selection pressure and differential fitness within populations.

### Funding

This work was supported by the Biotechnology and Biological Sciences Research Council, the Department for Environment, Food and Rural Affairs, the Natural Environment Research Council, the Scottish Government and the Wellcome Trust, under the Insect

Pollinators Initiative [grant number BBI0008281]. The funders had no role in study design, data collection and analysis, decision to publish, or preparation of the manuscript.

## Grant Disclosures

The following grant information was disclosed by the authors:
Biotechnology and Biological Sciences Research Council.
Department for Environment, Food and Rural Affairs.
Natural Environment Research Council.
Scottish Government and the Wellcome Trust.
Insect Pollinators Initiative: BBI0008281.

## Competing Interests

The authors declare there are no competing interests.

## Author Contributions

- Graham R. Wood conceived and designed the experiments, analyzed the data, contributed reagents/materials/analysis tools, wrote the paper, prepared figures and/or tables.
- Nigel J. Burroughs and David J. Evans conceived and designed the experiments, contributed reagents/materials/analysis tools, reviewed drafts of the paper.
- Eugene V. Ryabov conceived and designed the experiments, performed the experiments, analyzed the data, contributed reagents/materials/analysis tools, reviewed drafts of the paper.

## Data Deposition

The following information was supplied regarding the deposition of related data:
EBI SRA study accession PRJEB5249, ERS395182; EBI SRA study accession PRJEB5249, ERS395180

NCBI Accession Nos. HM067437 and HM067438.

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
