# Peer review of "Error correction and diversity analysis of population mixtures determined by NGS"

_PeerJ, doi:10.7717/peerj.645_

## Round 0.1 · original submission · Major Revisions

· Academic Editor

Major Revisions

This manuscript describes a new method for correcting errors introduced by next generation sequencing methods using the nucleotide distribution. This has potential but it is recommended to test the accuracy of the method. Also please revise the manuscript as suggested by two reviewers.

·

Basic reporting

No Comments. The article is well written.

Experimental design

The authors develop a new method for correcting errors introduced by next generation sequencing (NGS) methods. In contrast to previous approaches, the method works at the level of the nucleotide distribution rather than the level of individual nucleotides. The method is first calibrated by measuring NGS error of a sequence with known distribution and then applied to correct nucleotide distributions from NGS samples.

The authors first sampled an accurately known mix of two viral recombinants with Illumina sequencing and excluded mismatches between the recombinants to provide a reference where all diversity was due to NGS. They calibrated their method on this data.

The authors then compared the viral population mix from low DWV level and high DWV level honeybees using their NGS error correction. Specifically, they compare a single site in the helicase region of deformed wing virus and the average across the capsid region.

This experimental design is problematic. No explanation is given for using these specific examples. If this is intended to be a general method of error correction, then each region of the deformed wing virus (and other viruses if possible) should be tested to avoid selection bias. Moreover this design provides no estimate of the error involved in the method or comparison with other approaches. The method accurately recovers the calibration dataset but this does not necessarily imply that the method accurately recovers the test data; might NGS sequencing errors not differ from dataset to dataset? The authors should first test their method on a different known dataset (or perhaps a dataset sampled with both traditional and NGS methods) to establish accuracy before jumping to new data and making inferences.

Validity of the findings

p-values are often reported as "negligible" and should be included (pages 10,11).

The authors note that "The highly significant difference between the Varroa-free and Varroa-infested nucleotide diversities are evident," (page 11) but do not note that this is true for the corrected and uncorrected data. As such, these results do not show that their error correction provides new insight. The corrected results are clearly significantly different but, as mentioned above, the experimental design does not establish the accuracy of their method. Low-diversity samples were affected by correction much more than high-diversity samples and the authors should establish that this is plausible rather than bias.

Additional comments

This is an interesting idea for error-correction where the end goal is diversity comparison rather than the actual sequence. My main worry is that the accuracy of the method was not tested and that the calibration step may introduce bias. The authors must establish the accuracy of their method on a known dataset; the use of a known dataset for the calibration step does not remove this requirement and simply adds to the number of datasets required, as with any calibration step.

Reviewer 2 ·

Basic reporting

No Comments

Experimental design

No Comments

Validity of the findings

No Comments

Additional comments

This manuscript describes a novel method to correct next-generation sequencing (NGS) errors in diversity analysis of viral populations. The method is based on an evolutionary model, the Jukes-Cantor model, and corrects NGS errors in the nucleotide distribution. The authors also implemented the method in a freely available software package. Overall the manuscript is well-written and the sequencing error correction method seems very useful in viral nucleotide diversity analyses. I only have several minor concerns:
1) Page 8: the formula says “Z0.95” while the text says “where z0.975 = 1:96 is the 97.5th percentile of the standard normal distribution”. This seems not consistent.
2) Page 10, Table 2: the last line should be “0 0 1 0 0” instead of “1 0 0 0 0”.

---

## Round 0.2 · accepted · Accept

· Academic Editor

Accept

These are adequate revisions as I have checked them all and the reviewer's requests have been satisfied. Hence I accept this paper. Congratulations to authors.

Reviewer 1 ·

Basic reporting

The paper is well written and contains appropriate background, explanation and figures.

Experimental design

The authors have included the results of calibrating on one dataset and testing another, and cross-validated partitions of their capsid genome tests. They have fully addressed my previous concerns with their experimental design.

Validity of the findings

The authors now present a theoretical justification for their method. Along with their separated dataset test and cross-validation, this justifies the validity of their findings.

Additional comments

The authors have addressed all of my concerns and i recommend the article be accepted.